# Could Targeting NPM1c+ Misfolding Be a Promising Strategy for Combating Acute Myeloid Leukemia?

**DOI:** 10.3390/ijms25020811

**Published:** 2024-01-09

**Authors:** Daniele Florio, Daniela Marasco

**Affiliations:** Department of Pharmacy, University of Naples “Federico II”, 80131 Naples, Italy; daniele.florio@unina.it

**Keywords:** nucleophosmin 1, AML–NPM1 mutations, NPM1c+ targeted therapies

## Abstract

Acute myeloid leukemia (AML) is a heterogeneous group of diseases classified into various types on the basis of distinct features concerning the morphology, cytochemistry and cytogenesis of leukemic cells. Among the different subtypes, the group “AML with gene mutations” includes the variations of the gene of the multifunctional protein nucleophosmin 1 (NPM1). These mutations are the most frequent (~30–35% of AML adult patients and less in pediatric ones) and occur predominantly in the C-terminal domain (CTD) of NPM1. The most important mutation is the insertion at W288, which determines the frame shift W288Cfs12/Ffs12/Lfs*12 and leads to the addition of 2–12 amino acids, which hamper the correct folding of NPM1. This mutation leads to the loss of the nuclear localization signal (NoLS) and to aberrant cytoplasmic localization, denoted as NPM1c+. Many investigations demonstrated that interfering with the cellular location and oligomerization status of NPM1 can influence its biological functions, including the proper buildup of the nucleolus, and therapeutic strategies have been proposed to target NPM1c+, particularly the use of drugs able to re-direct NPM1 localization. Our studies unveiled a direct link between AML mutations and the neat amyloidogenic character of the CTDs of NPM1c+. Herein, with the aim of exploiting these conformational features, novel therapeutic strategies are proposed that rely on the induction of the selective self-cytotoxicity of leukemic blasts by focusing on agents such as peptides, peptoids or small molecules able to enhance amyloid aggregation and targeting selectively AML–NPM1c+ mutations.

## 1. Introduction

Acute myeloid leukemia (AML) is a prevalent form of acute leukemia in adults, with a cure rate of 40–50% among individuals aged 18–60 years when treated with standard chemotherapy and/or allogeneic stem cell transplantation. However, the incidence of AML rises significantly with age, typically affecting individuals around 70 years old, and the cure rate drops dramatically to 5–10% in older adults [1]. Despite advancements in supportive care and relapse prediction, a high cure rate in AML remains elusive, and to address this medical challenge, it is crucial to deepen the cellular and molecular mechanisms at the basis of the distinct AML subtypes. Genetic heterogeneity is a critical factor in predicting treatment response and developing therapeutic strategies [2]. To actually tailor patients’ care, concomitant genetic mutations are also delineated, pointing out the importance of individualized assessment to optimize the efficacy of treatment protocols [3,4].

Genomic studies, focused on distinct gene sequences and mutations, unraveled the great molecular heterogeneity in AML patients [2]: in ~30–35% of cases, the most prevalent genetic mutations involve nucleophosmin 1 (NPM1), occurring in approximately one-third of newly diagnosed cases across age groups [5]. This subtype of AML, known as “AML with mutated NPM1”, was identified in 2005 through unique immunohistochemical patterns fully predictive of NPM1 mutations [5,6]. Indeed, in AML it exhibits peculiar molecular and clinical characteristics: an abnormal cytoplasmic displacement of its mobility to drive clonal hematopoiesis, its exclusion with recurrent cytogenetic abnormalities, distinctive gene expression and micro-RNA profiles. For these features, the World Health Organization (WHO), in 2017, recognized “AML with mutated NPM1” as a separate entity among lymphohematopoietic malignancies [7,8].

This review will discuss NPM1-mutated forms of AML, detailing the structural and functional consequences of these mutations.

## 2. Structure and Functions of NPM1

NPM1 is a multifunctional protein with nuclear chaperone functions [9]. It is mainly present in nucleoli, where it takes part in rRNA maturation processes [10] and embryonic development [11]. It is directly implicated in primitive hematopoiesis and hematopoietic malignancies as myelodysplastic syndrome (MDS), since NPM1 plays a critical role in the maintenance of hematopoietic stem cells (HSCs) in preserving the functional integrity of these cells in the context of competitive transplantation (HSCT) [12] and the transformation of MDS into leukemia as well as in the modulation of gene expression and signaling pathways that govern cell survival [13]. NPM1 is observed in different cellular compartments in response to various types of cellular stresses [14,15], and it is found to be overexpressed or mutated in various types of tumors, including gastric, ovarian, bladder and prostate carcinomas and in hematological malignancies [16,17]. NPM1 has a modular structure, constituted by three domains, as schematically described in Figure 1 [18]. The N-terminal domain (NTD, residues ~1–117), is highly conserved within the *nucleoplasmin* family and serves as an oligomerization core region enabling self- and hetero-oligomerization with other proteins. Based on X-ray studies, the structure of human NPM1-NTD displays a unique arrangement of eight β-barrels, resembling a jelly roll barrel [19]: in it, NPM1 monomers assemble as donut-shaped homo-pentamers, exhibiting an asymmetric distribution of charges, with clustered negatively charged residues on the top surface of the oligomer. These pentamers interact in a head-to-head fashion, forming a decamer that exhibits an adaptable structural plasticity at the pentamer–pentamer interface [19]. When in oligomeric status, NPM1 engages in interactions with many proteins, including protamines, protamine-like proteins, histones, nucleosomes, DNA repair proteins like XRCC1 (X-ray repair cross-complementing protein 1), transcription factors, and APE1 (apurinic/apyrimidinic endoribonuclease 1) [20]. Post-translational modifications dynamically regulate this multimeric state; indeed, phosphorylation regulates the monomer–pentamer equilibrium by inducing the disassembly of the pentamer into unstable and unfolded monomers, and this polymorphism regulates NPM1 cellular localization and function [21]. The oligomeric form of NPM1 is associated with its nucleolar localization and cellular proliferation; instead, the monomer is associated with responses to DNA damage and apoptosis [22]. The NTD contains a nuclear export signal (NES), which further confirms its regulatory role in nucleo-cytoplasmic shuttling [23].

The central portion of NPM1 is an IDR (intrinsically disordered region) characterized by two acidic domains rich in aspartic and glutamic acids (regions 119–133 and 161–188) [24]. Their negative charges can mimic the DNA and RNA sequences, allowing binding to histones H1, H3, H4, H2A and H2B [25,26]. Within this region, there is a specific sequence serving as a nuclear localization signal (NLS) [27], which has been demonstrated to drive the liquid–liquid phase separation (LLPS) of NPM1 within the nucleolus [22,28]. This process occurs through distinct mechanisms: two heterotypic, involving nucleolar proteins such as NCL (nucleolin) and FBL (fibrillarin) that display R-rich motifs or nascent ribosomal RNA (rRNA), and one homotypic, based on internal interactions mediated by electrostatic forces within the polyampholytic IDR. These interactions maintain the liquid-like state of the nucleolar granular component (GC) during ribosome assembly [21]. Indeed, heterotypic NPM1-mediated LLPS mechanisms initiate the early stages of ribosome assembly, close to the dense fibrillar component (DFC). During this phase, ribosomal proteins and rRNAs integrate into pre-ribosomal particles and mask sites of interaction with NPM1, reducing the affinity for it [29]. Subsequently, NPM1 homotypic mechanisms take place, guiding the pre-ribosomal particles for the exit from the nucleolus [30].

Mutagenesis studies revealed that deletions within the NTD or central region impair chaperone activity: NTD deletions such as DN35 and DN90 showed 84% and 66% reduced activity and the further deletion of 30 additional amino acids (DN119) drastically reduced this activity to approximately 10% compared to the full-length protein. Similarly, deletions in the IDR region (NPM1.3 and NPM1.1-ΔA1A2) induce alterations in the protein dynamics and dysregulation within NPM1’s interactome, hampering its chaperone activity [24,31].

The C-terminal domain (CTD) (residues 225–294) is a globular region, in wt form, containing a three-helix bundle [32] in which helices H1 and H3 are almost coaxial with the opposite polarities, whereas the connecting helix, H2, is tilted by ~45° with respect to the other helices [32]. The overall structure is stabilized by a small hydrophobic core among four aromatic residues, Phe268, Phe276, Trp288 and Trp290, which are also involved in the uncommon nucleolar localization signal (NoLS) that regulates also NPM1’s interactions with ribosomal DNA within the nucleolus [33]. This domain is mainly involved in DNA/RNA recognition and regulation [34].

## 3. *NPM1* Is the Most Commonly Mutated Gene in Adult AML

### 3.1. Common NPM1 Mutations and Structural Consequences

NPM1 mutations are crucial markers of AML [35]. They occur in the context of preleukemic clonal hematopoiesis [36] and are prognostically favorable in the absence of FLT3-ITD mutations [37]. RT-PCR and direct sequencing studies have revealed that the majority of mutations in the *NPM1* gene predominantly occur in exon 12. Within this exon, researchers have identified about 50 different types of mutations, specifically 14 distinct mutations [5] were identified and are reported in Table 1, including the common A–F mutations and another eight variants, named from J to Q [38]. Type A is the most frequent mutation (75–80% of cases) and consists of a duplication/insertion of a TCTG tetranucleotide between nucleotide positions 960 and 961. The resulting shift in the reading frame involves alterations in the CTD by replacing the last seven amino acids (WQWRKSL) with eleven different residues (CLAVEEVSLRK). Type B, the second most common mutation (10% of cases), presents an insertion of four base pairs (CATG) at the same nucleotide position as type A, while the D mutation has a similar insertion of four base pairs of CATG and provides identical aminoacidic mutations. In types E and F, nucleotides from 965 to 969 (GGAGG) are deleted and two different 9 bp sequences are inserted, causing a distinct frame shift in nine amino acids [39]. As the fourth most common mutation, type I (c.863_864insCTTG) was identified in 1.7% of 877 patients [40]. All these mutations are heterozygous and retain a wt allele, and they cause the loss of one Trp (E-F) or, more commonly, two Trp residues (A-D) [41] and the gain of a new NES [42].

The most frequent NES motif (L-xxx-V-xx-V-x-L) is present in 7 of the 14 leukemic mutants, while the other mutants exhibit NES with different hydrophobic amino acids substituting Val (Phe in O- and Met in Q-type mutations) (Table 1). Hence, the mutations shift the balance toward nuclear export, causing cytoplasmic localization of the protein, NPM1c+ (Figure 2) [43], which interacts with XPO1 (Exportin 1) through the NES motif, which is responsible for the transfer of NES proteins from the nucleus to the cytoplasm [44]. All the observed exon 12 AML mutations cause the loss of the three-helix bundle in the CTD [32].

### 3.2. Rare Mutations of Exon 5 and Fusion Transcripts of NPM1c+

While almost all the mutations involve exon 12, rarely (<1%) mutations can concern exons 5 [45], 9 and 11 [46], leading to NPM1 cytoplasmic mislocation. In this context, a recent study involving three patients, namely, PG patients 1, 2 and 3, was focused on exon 5 mutations (Table 2): PG 1 with an in-frame insertion of 21 nucleotides at c408–409 that produces a mutant 7 amino acids longer than the wt protein (p.L136_137insAEDVKLL), PG 2 with an out-of-frame insertion of 18 nucleotides at c409–410, leading to a truncated protein of 137 aa (p.S137_K137fs*) and PG 3, with an in-frame insertion of 27 nucleotides at c424–425, leading to a mutant protein having 9 more amino acids than the wt (p.K141_142insLSALSISGK). Another study reported on a fourth case of exon 5 mutation, patient MLL 4, in which in-frame insertions/duplications, each consisting of nine nucleotides, occurred at c399–400 and c400–401, leading to a six amino acid protein longer than the wt (p.K134_Q134insLLSGLQ). Immunohistochemistry and analysis of fusion transcripts such as GFP-NPM1 mutants in NIH-3T3 cells highlighted that all the exon 5 mutations do not lead to the loss of NoLS but, nevertheless, exhibit a cytoplasmic localization for the presence of an additional NES. Indeed, with the exception of PG 2, all the other mutants retained the wt CTD [45].

Very recently, a mutation called the NPM1_MutSong mutant (from the name of the patient Song), in exon 5 was discovered, generating an additional NES (LALELGNLSI) (Table 2) in the central region which caused its cytoplasmatic localization despite the preservation of the Trp^288^ and Trp^290^ of the NoLS [47].

### 3.3. Consequences of Cytoplasmic Mislocation of NPM1c+

A key characteristic of NPM1c+ in AML patients is the high expressions of the homeobox (HOX) transcription factors (TFs), including the *HOXA/B* cluster genes, *MEIS1* (myeloid ecotropic viral integration site 1) and *PBX3* (pre-B-cell leukemia homeobox 3) [48]. Activation of these TFs leads to an increase in the self-renewal of leukemic clones [44], therefore inhibiting the expression of these TFs through Menin-inhibitors or targeting XPO1 to prohibit NPM1c+ translocation is currently in clinical investigation (NCT04067336; NCT02667873) [49,50].

AML with mutations in the *NPM1* gene morphologically displays the features of myeloblastic leukemia with or without differentiation (predominant blast cells, cytoplasmic granules, irregular nuclei and heterogeneous subtypes) and those of acute leukemia with monocytic/monoblastic differentiation (undifferentiated cells, differentiation based on monocytic-lineage blast cells, prominent blast morphology). Blast cells often exhibit cup-like nuclei [51]. Immunophenotypically, cells with NPM1 mutations are frequently positive for CD33, CD117 and MPO (myeloperoxidase) in myeloblastic-like cases [52]. A majority of AML cases have CD34-negative blasts, while approximately one-third lack HLA-DR (human leukocyte antigen-DR isotype) expression [53].

The NPM1 locus is involved in translocations associated with hematologic malignancies, including (i) acute promyelocytic leukemia t(5;17)(q35;q12) leading to NPM1-RARα (retinoic acid receptor alpha fusion protein), (ii) anaplastic large cell lymphoma t(2;5)(p23;q35) leading to NPM1-ALK (anaplastic lymphoma kinase fusion protein), and (iii) myeloid neoplasms t(3;5)(q25;q35) leading to NPM1-MLF1 (myeloid leukemia factor 1 fusion protein) [54,55]. The fusion protein NPM-ALK resulted in a significant percentage of advanced anaplastic large cell lymphoma cases and immunohistochemical assays indicated a cytoplasmic localization [56].

In general, the cytoplasmic mislocation of AML-NPM1 dysregulates several cellular processes, leading to uncontrolled centrosome duplication, the inhibition of tumor suppressor genes, the activation of caspase 6 and 8 proteolytic activities, impairment of the DNA repair pathways and the activation of the Myc oncogene [57]. These imbalances can drive myeloproliferation and the development of leukemia. However, animal studies revealed that these imbalances alone may not always lead to full-fledged disease, indicating the requirement for NPM1 mutations and cooperating mutations in other genes, like DNA methyltransferase 3A (DNMT3A), to allow the progression to overt leukemia [58].

Moreover, in AML, the total absence of nucleolar NPM1 is not frequent since AML mutations are always heterozygous due to the non-viability of homozygous NPM1 mutations. This effect is reflected in the formation of heterodimers among mutated and wt forms of NPM1, locating them to the cytoplasm [59]. The molecular mechanism underlying the induction of the leukemic state in myeloid cells remains a critical issue. One key aspect involves the upregulation of Class I *HOX* genes, which are crucial for proliferation, differentiation and self-renewal in hematopoietic stem cells, and the upregulation of *HOX-A* and -*B* genes, linked to a stem-cell-like state, was observed in AML-NPM1c+ cases [60]. NPM1c+ might contribute to the leukemic phenotype by transporting other molecules into the cytoplasm, altering their functions [56], as demonstrated for the transcription factor PU.1/spi-1 (Spi-1 proto-oncogene), which resulted in impaired differentiation and the monocytic features of AML cells [61]. Similarly the bromo-domain protein 4 (BRD4) is an epigenetic regulator of transcription that is usually inhibited by wt NPM1, and the reduction of this inhibition due to cytoplasmatic localization led to an increase in the transcription of several related genes such as the anti-apoptotic *Bcl-2* and oncogene c-*MYC* [62].

## 4. NPM1c+ Mutations Govern the Amyloidogenicity of the CTD

Despite numerous studies of the cellular effects of the cytoplasmic localization of NPM1, few examples of research addressing the conformational effects of AML mutations were available up to 10 years ago. One explanation could concern the very low yields in attempts concerning the protein expression of mutated forms of CTD-NPM1c+. For this, we are conducting studies on the structural consequences of AML mutations using synthetic polypeptides [63,64]. In these investigations, biophysical approaches demonstrated, unequivocally, that an amyloid-aggregation propensity is a direct consequence of AML mutations [63,65,66,67,68,69,70,71,72,73,74,75,76]. Initial studies, carried out by performing a protein dissection of the CTD, were focused on the analysis of peptides covering the three-helices of the bundle, H1, H2 and H3. In detail, in the H2 region, NPM1_264–277_, exhibited a remarkable tendency to form amyloid-like assemblies with fibrillar morphology and β-sheet structure under physiological conditions. These assemblies proved to be cytotoxic in SH-SY5Y neuroblastoma cells. A systematic alanine scanning study of the 264−277 region indicated as the “basic amyloidogenic unit” the stretch FINYV (268−272) [65] in the initial stages of the self-recognition process [66]. These novel findings led us to a hypothesize that the structural destabilization of the CTD in NPM1 mutations could facilitate aggregation by exposing the amyloidogenic H2 region. To probe this hypothesis, similar studies were carried out on peptides spanning the third helix, H3, both in wt and AML variants [67,76]. NMR studies revealed that the H3 wt has a prevalent α-helical structure, while the mutants, type A and E, exhibited greater flexibility in promoting amyloid aggregation. Subsequently, all the common mutations in exon 12, were assessed to form amyloid aggregates, displaying variable kinetics and levels of oligomerization [76]. By assuming an aggregomic perspective on the structural consequences of AML mutations, the conformational and aggregative behavior of the entire CTD were investigated in wt, mutA, mutF and mutC variants [70,74,77].

These structure–activity relationship (SARs) studies introduced novel insights into the possible molecular mechanisms associated with AML onset and progression and prompted further exploration of the structural determinants that regulate the aggregation process [78,79].

## 5. Therapeutic Strategies Targeting NPM1c+

Numerous therapeutic approaches are currently being explored to target NPM1. As schematically described in Figure 3, they include: (1) the disruption of its oligomerization, (2) the modulation of nucleolar assembly, (3) the inhibition of translocation, (4) the induction of nucleolar starvation and (5) the promotion of its degradation via the proteasomal pathway [8,16,27,43].

Despite significant progress in AML treatments, ~50% of patients with NPM1 mutations still do not respond to current therapies and succumb to the disease [80]. These data highlight the urgent need for innovative approaches to improve treatment outcomes.
Figure 3Scheme of therapeutic strategies targeting the structure (red panel), levels (blue panel) and localization (green panel) of NPM1 in AML. This figure is adapted from [81].
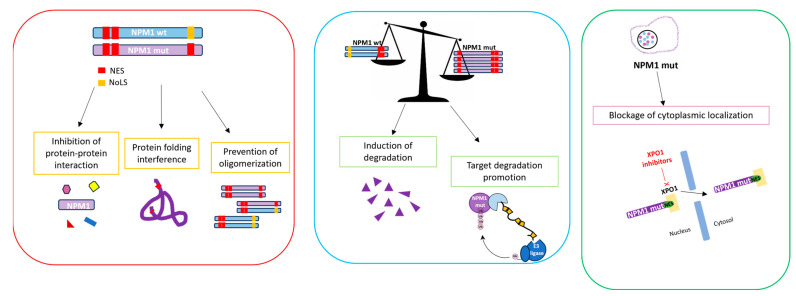



In leukemic cells bearing NPM1c+, the nucleolus becomes particularly susceptible to stress due to the partial depletion of NPM1 caused by both haploinsufficiency and mislocalization [82], thus the induction of nucleolar stress has emerged as a potential therapeutic strategy for NPM1 AML.

### 5.1. Therapeutics Targeting NPM1 Protein–Protein Interactions

In the field of small molecules as potential therapeutics, NSC348884 (Figure 4) was initially reported to prevent the formation of NPM1 oligomers [83], but its MOA is still debated. Indeed, several pieces of experimental evidence indicated that NSC348884 was able to interfere with NPM1 oligomerization in vitro by targeting the hydrophobic NTD, leading to the unfolding of the tertiary structure and thus altering its interactome; furthermore, along with inhibitory effects on proliferation and pro-apoptotic effect in OCI cells [84]. On the other hand, for other authors, NSC348884 did not affect the NPM oligomer in many leukemia cells and the cell sensitivity to NSC348884 treatment was not potentiated by AML-associated NPM mutation. An unknown effect of NSC48884 on the cell-surface adhesion, could play a key role in the complex cellular response to the NSC48884 treatment [85].

Other compounds can target the CTD of NPM1c+, for example, avrainvillamide (Figure 4), is a prenylated indole alkaloid, highly oxidized, isolated from the marine fungus *Aspergillus* sp. *CNC358*. It not only demonstrated in vitro the ability to form a covalent linkage with Cys^275^ in certain AML NPM1 mutants (types A and E) but also inhibited the nuclear export of the XPO1 protein, altering NPM1’s cellular localization and partially relocating NPM1c+ to the nucleus [86]. Avrainvillamide exhibited greater efficacy in anti-proliferative activity in OCI-AML3 than in primary AML cells, likely due to the unfolded state of the CTD in mutated proteins. Additionally, it induced proteasomal degradation of NPM1c+ and promoted differentiation of OCI-AML3 cells [87].

### 5.2. Therapeutics Targeting the Nucleolus of NPM1c+

Falini and collaborators reported on a phase 2 pilot study (2014-000693-182; 2014-003490-412) investigating the safety and efficacy of a single-agent, dactinomycin (Figure 4), which is a well-known antibiotic from the actinomycin group that exhibits high antibacterial and antitumor activity. It is a natural polypeptide isolated from soil bacteria of *Streptomyces parvullus* [88]. This drug can induce complete remission associated with nucleolar stress response in relapsed/refractory NPM1c+ adult patients and is relatively well tolerated [89]. Several studies reported that NPM1c+ disrupts the formation of nuclear bodies (NBs) [90], which are key regulators of mitochondrial fitness and senescence; thus, AML cells harboring NPM1c+ often experience impaired mitochondrial function [91]. Actinomycin D intervenes in this scenario by influencing mitochondria, releasing mDNA, activating cyclic GMP-AMP synthase signaling, generating reactive oxygen species (ROS) and restoring NB formation, which, in turn, triggers TP53 activation and senescence [90]. Conversely, another approach involves the combination of arsenic trioxide (ATO) with all-trans retinoic acid (ATRA) (Figure 4). ATO is very potent old drug reintroduced into new medicine by Chinese studies on acute promyelocytic leukemia (APL) treatment and rapidly approved by the FDA for relapsed cases [92], while ATRA is the most active metabolite of vitamin A with significant anticancer properties [93] (Figure 4). Their synergistic effect triggered a proteasome-dependent degradation of NPM1c+, leading to cell death in OCI-AML3 cells and primary AML cells harboring NPM1 mutations [90,94]. Other compounds, like deguelin (Figure 4), which is a naturally occurring flavonoid with anti-cancer activities [95], and (−)-epigallocatechin-3-gallate (EGCG), which is the major catechin found in green tea [96], demonstrated the ability to reduce NPM1mut expression levels and to induce apoptosis through the upregulation of caspase 6 and caspase 8 in both OCI-AML3 and IMS-M2 cells [97,98]. EAPB0503 (1-(3-methoxyphenyl)-*N*-methylimidazo[1,2-a]quinoxalin-4-amine) is an imidazoquinoxaline derivative [99] able to induce proteasome degradation mediated by SUMOylation and ubiquitylation of NPM1c+. The mechanism implies the restoration of the NPM1wt within the nucleolus, the induction of apoptosis upon the downregulation of HDM2 and the activation of p53, and a reduction in the leukemia burden in NPM1c+ AML xenografts [100,101]. PROTACs (proteolysis-targeting chimeras) are heterobifunctional molecules consisting of one ligand that binds to a protein of interest (POI) and another that can recruit an E3 ubiquitin ligase [102] (Figure 4). They result in attractive therapeutics since they are able to trigger ubiquitination and proteasome-mediated degradation of proteins [103], and they have been employed in promoting the degradation of fused oncoproteins in MLL-PTD leukemia subtypes [104], suggesting its application could be potentially extend to NPM1c+. Specifically, its selective degradation using a degron-tag could lead to the differentiation and growth arrest of NPM1c+ cell lines (OCI-AML3 and IMS-M2) [44], confirming the effectiveness of this approach.

### 5.3. Therapeutics Targeting NPM1c+ Localization

Given the characteristic alteration in the shuttling of NPM1 between the nucleus and the cytoplasm in AML, numerous strategies employ nuclear export inhibitors, such as leptomycin B (Figure 4). Leptomycin B (LMB) is an antifungal antibiotic from *Streptomyces* species, and it is a specific inhibitor of nuclear export protein XPO1 [105]. Unfortunately, it is associated with severe toxicity due to its irreversible binding mechanism to XPO1 [106]. More recently, a class of reversible inhibitors known as selective inhibitors of nuclear export (SINEs) was developed [107], and among them, KPT-330 (selinexor) [108] and KPT-8602 (eltanexor) (Figure 4) [109] showed anti-leukemic activity by inhibiting the interaction between NPM1-mut and XPO1, improving the tolerability and also synergistic activity with BCL-2 inhibitors by increasing apoptosis in NPM1-AML cells [110], as currently being evaluated in early phase trials (NCT03955783).

### 5.4. Therapeutics as Menin Inhibitors

Menin inhibitors are emerging as promising therapeutics in NPM1c+-AML. The menin protein is a crucial regulator of tissue-specific gene expression [111] and a co-factor for histone-lysine-*N*-methyltransferase 2A (KMT2A), influencing histone methylation and correlating with active transcription of *HOX* genes. The aberrant expression of *HOXA* and *HOXB* genes and of their co-factor MEIS1 is a hallmark of NPM1-mutated AML [112]. The abnormal cytoplasmic localization of NPM1c+ directly influences *HOX* gene expression [44], indicating a link between cellular localization and gene regulation. In this context, the inhibition of the KMT2A/menin complex by chemical agents, such as VTP-50469 and MI-3454, demonstrated significant anti-leukemic activity, reducing cell proliferation, downregulating *HOXA/B* clusters and *MEIS1* gene expression, promoting a marked differentiation of leukemic cells, and reducing the AML engraftment and survival in mouse patient-derived xenograft (PDX) models [113,114].

Ongoing clinical trials concerning menin inhibitors are producing important preliminary results in terms of safety and efficacy for patients with NPM1-mutated or KMT2A-rearranged AML [115].

KO-539 (Figure 5) induces complete remission (CR) in two out of six patients with relapsed/refractory acute myeloid leukemia (R/R AML), including a minimal residual disease (MRD)-negative CR in a NPM1-mutated AML patient co-mutated for DNMT3A and KMT2D [115]. Actually, a trial enrolling NPM1-mutated and KTM2A-rearranged AML patients at different doses (NCT04067336) is ongoing. SNDX-5613 (Figure 5) is an analog of VTP-50469, which demonstrated significant activity in mouse models of NPM1-mutated AML, with some animals remaining in CR one year after treatment cessation [114,116]. A related trial in adult patients with R/R NPM1-mutated or KTM2A-rearranged AML showed a 29% overall response rate and RNA-Seq analysis indicated downregulation of *MEIS* and *HOXA9* genes and upregulation of differentiated antigens CD11b, CD14, and CD13 and actually has gone into phase 2 (NCT04065399). A recent study highlighted the efficacy of JNJ-75276617 (with a patented structure) with a significant preclinical activity against OCI-AML3 cell lines and primary AML cells by disrupting the menin/KMT2A protein complex and the resulting data support the first-in-human study to evaluate JNJ-75276617 as monotherapy for patients with R/R AML with KMT2A or NPM1 alterations (NCT04811560) [117]. BMF-219 (Figure 5) is a selective and orally bioavailable irreversible inhibitor able to abrogate menin-dependent oncogenic signaling, with antiproliferative effects on menin-dependent AML and diffuse large B-cell lymphoma (DLBCL) cell lines [118]. The safety, tolerability and clinical activity of escalating doses of BMF-219, orally administered once daily, are currently being assessed in a phase I study (NCT05153330) [119]. DSP-5336 (Figure 5) is an effective and selective inhibitor, specifically capable of inhibiting the growth of leukemic cell lines, such as MV-4-11, MOLM-13, KOPN-8, and OCI-AML3, harboring MLL-r (mixed-lineage leukemia-rearranged) or NPM1 mutations but not significantly affecting those without such mutations [120]. At a molecular level, it positively modulated menin-regulated gene expression, leading to a reduction in *MEIS1* and *HOXA9*. Moreover, in both human and murine leukemia models, the compound demonstrated the inhibition of blast colony formation and induced CR [120]. A phase 1/2 dose escalation/dose expansion study of DSP 5336 in patients with R/R AML is ongoing (NCT04988555).

### 5.5. Therapeutics Targeting Aggregation of NPM1c+

On the basis of our studies, we are proposing an alternative and innovative therapeutic route by exploiting the molecular aggregation exhibited by AML’s mutated forms. The amyloid self-recognition can likely concur with the formation of cellular aggregates; thus, the use of external agents able to promote amyloid aggregation could be a valuable tool for inducing cytotoxicity. In general, the promotion of the aggregation of specific proteins allows them to disrupt their normal cellular functions or induce cytotoxicity in target cells [121]. Recently, we followed this approach by employing two small molecules, phenoxazine compounds derivatives of orcein: dihydroquinazoline, named smA and hexahydroquinoline, smB (Figure 4), which were already demonstrated to be accelerators of amyloid aggregation. These compounds showed a significant influence on the aggregation mechanism of NPM1c+. In particular, smB specifically induced cytotoxicity in OCI-AML3 cell lines overexpressing NPM1c+, thereby promoting the formation of intracellular amyloid aggregates [122]. This discovery marks a significant milestone, as it represents the first instance of agents targeting the molecular aggregation of NPM1 in AML mutations.

## 6. Conclusions

Given the plethora of cellular processes involving NPM1, many investigations unraveled its role in diseases, where it appeared upregulated and/or mutated. Very recently, NPM1-mutated MDS and chronic myelomonocytic leukemia CMML cases represent NPM1-mutated AML diagnosed at an early stage and the NPM1 mutant defines AML irrespective of the blast count [123]. The deep knowledge of cellular pathways allows us to pave the way for new targeted treatments that specifically aim at the mutated genetic pathways responsible for cancer cell growth. These advances have not only made it possible to create more effective and targeted therapies but have also highlighted the importance of personalized medicine, which provides customized treatments for the individual genetic profiles of patients. Important issues to be evaluated during the diagnose and treatment of NPM1-mutated AML are the NPM1 mutational status, the timing of HSCT, MRD monitoring and ELN (Elastin) genetic-based risk stratification [124].

The combination of multiple agents is a dominant trend for NPM1-mutated AML, such as venetoclax-based regimens and XPO1 inhibitors combinations. Interestingly, leukemic cells in the primitive subtype are more sensitive to certain kinase inhibitors and the addition of kinase inhibitors to the treatment might achieve benefits in this subtype of NPM1-mutated AML. These promising discoveries mark a significant step forward in improving the treatment prospects for AML patients. These findings indicate the complex ways in which cytoplasmic localization of NPM1mut can impact various cellular processes, ultimately contributing to the development of leukemia.

## Figures and Tables

**Figure 1 ijms-25-00811-f001:**
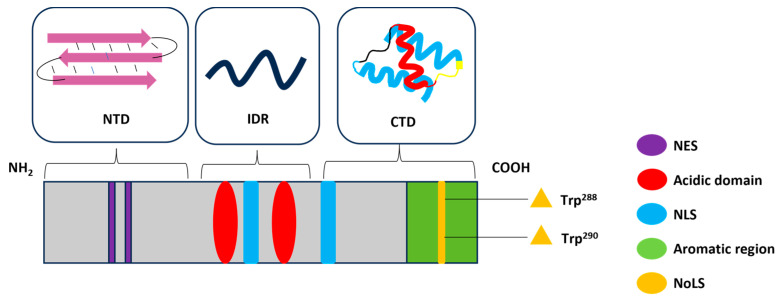
Structural and functional domains of the NPM1 protein. NTD (N-terminal domain); IDR (intrinsically disordered region); CTD (C-terminal domain); NES (nuclear export signal); NLS (nuclear localization signal); NoLS (nucleolar localization signal).

**Figure 2 ijms-25-00811-f002:**
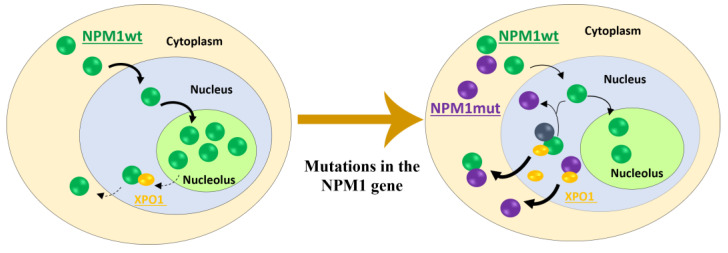
Schematic representation of the nucleo-cytoplasmic shuttling of NPM1wt and of NPM1-mut. (**Left**) Mechanism of nucleo-cytoplasmic shuttling of NPM1wt: the nuclear import of the protein (arrow) greatly predominates over the nuclear export (dotted arrow). Thus, NPM1wt mainly resides in the nucleolus. (**Right**) Aberrant cytoplasmic localization of both NPM1mut and NPM1wt in AML.

**Figure 4 ijms-25-00811-f004:**
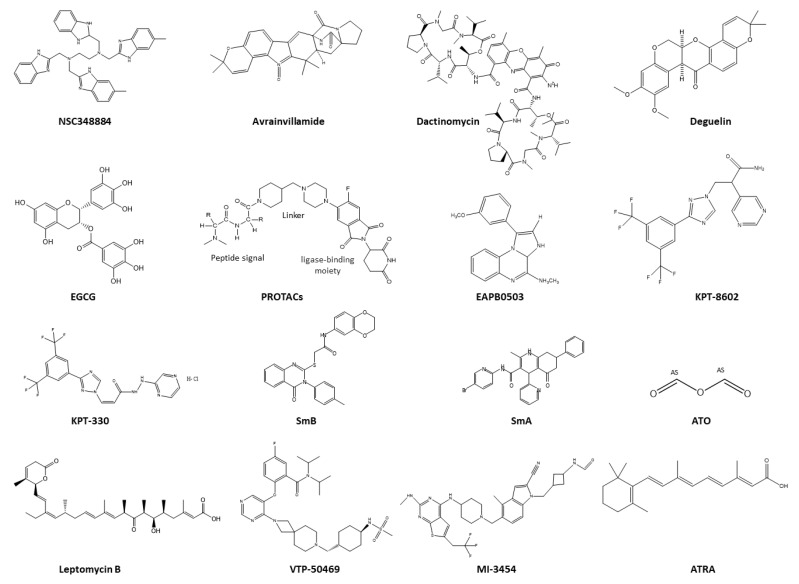
Chemical structures of the described small molecules as potential therapeutics in AML.

**Figure 5 ijms-25-00811-f005:**
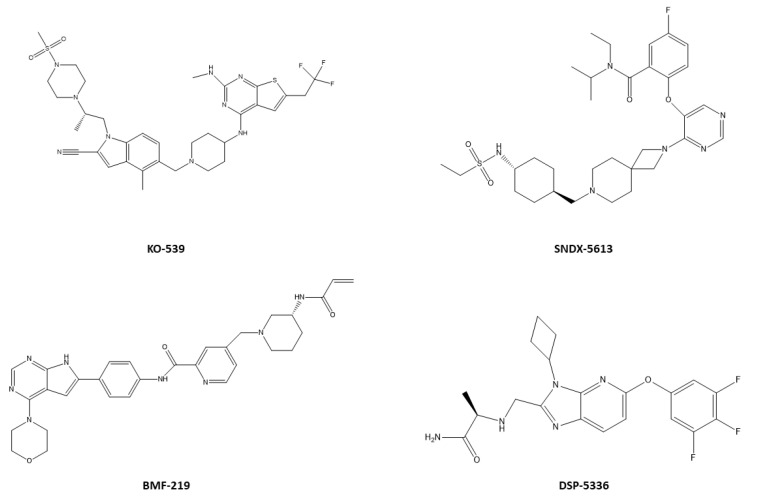
Chemical structures of menin inhibitors currently undergoing clinical evaluation.

**Table 1 ijms-25-00811-t001:** Common and rare mutations in exon 12 of the *NPM1* gene. In red, residues constituting new NES, and in blue, the stop signal. In green, the two tryptophans, Trp^288^ and Trp^290^, of the NoLS.

	Nucleotide Sequence	Protein
wt NPM1	GATCTCTG…GCAGT…GGAGGAAGTCTCTTTAAGAAAATAG	^286^DLWQWRKSL^294^
Common mutations	GATCTCTGTCTGGCAGT…GGAGGAAGTCTCTTTAAGAAAATAG	^286^DLCLAVEEVSLRK^298^
	GATCTCTGCATGGCAGT…GGAGGAAGTCTCTTTAAGAAAATAG	^286^DLCMAVEEVSLRK^298^
	GATCTCTGCGTGGCAGT…GGAGGAAGTCTCTTTAAGAAAATAG	^286^DLCVAVEEVSLRK^298^
	GATCTCTGCCTGGCAGT…GGAGGAAGTCTCTTTAAGAAAATAG	^286^DLCLAVEEVSLRK^298^
	GATCTCTG…GCAGTCTCTTGCCCAAGTCTCTTTAAGAAAATAG	^286^DLWQSLAQVSLRK^298^
	GATCTCTG…GCAGTCCCTGGAGAAAGTCTCTTTAAGAAAATAG	^286^DLWQSLEKVSLRK^298^
	GATCTCTG…GCAGTCTCTTTCTAAAGTCTCTTTAAGAAAATAG	^286^DLWQSLSKVSLRK^298^
	GATCTCTCCCGGGCAGT…AAGTCTCTTTAAGAAAATAG	^286^ DLSRAVEEVSLRK ^298^
	GATCTCTG…GCAGTCCCTTTCCAAAGTCTCTTTAAGAAAATAG	^286^ DL W QSLSKVSLRK ^298^
	GATCTCTGTAGCGCAGT…GGAGGAAGTCTCTTTAAGAAAATAG	^286^ DLCTAVEEVSLRK ^298^
	GATCTCTGCCACGCAGT…GGAGGAAGTCTCTTTAAGAAAATAG	^286^ DLCHAVEEVSLRK ^298^
	GATCTCTGGCAGCGTTTCCAGGAAGTCTCTTTAAGAAAATAG	^286^ DL W QRFQEVSLRK ^298^
	GATCTCTGTACCTTCCT…GGAGGAAGTCTCTTTAAGAAAATAG	^286^ DLCTFLEEVSLRK ^298^
	GATCTCTG…GCAGAGGATGGAGGAAGTCTCTTTAAGAAAATAG	^286^ DL W QRMEEVSLRK ^298^
	**Nucleotide Change**	
Rare mutations	c.864_876delinsTCGGAGTCTCGGCGGAC	^286^DLCRSLGGLSLRKA^299^
	c.864_873delinsTCAAGACTTTCTTA	^286^DLCQDFLKVSLRKA^299^
	c.867_875delinsAGATTTCTTAAATC	^286^DLWQDFLNRLFKRIVA^301^
	c.868_876delinsGGGATAGCGATGC	^286^DLWQGIAMLSLRKA^299^
	c.868_876delinsGGGGTGGGGAATC	^286^DLWQGVGNLSLRKA^299^
	c.863_871delinsCGACCCTCCTGGG	^286^DLSTLLGEVSLRKA^299^

**Table 2 ijms-25-00811-t002:** Mutations in exon 5 of the *NPM1* gene. In red, residues changed in the protein sequence with respect to the wt, and in green, the two tryptophans, Trp^288^ and Trp^290^, of the NoLS.

	In-Frame Insertion/Duplications	Protein
Exon 5 Mutations	c408–409 (F,5′-GCGGAGGATGTGAAACTCTTA)	DVKLL^136^AEDVKLL…^286^DLWQWRKSL^294^
	c409–410 (F,5′-AATGATCTGTCACTTCTG)	DVKLL^137^K
	c424–425 (F,5′-TTTCTGCCTTAAGTATATCTGGAAAGC)	ISGK^141^LSALSISGK…^286^DLWQWRKSL^294^
	c399–400 (F,5′-CAACTCTTA) and c400–401 (F,5′-GTGGGCTGC)	EEDV^134^QLLSGLQ…^286^DLWQWRKSL^294^
	c406–423 (F,5′-GCCCTGGAACTGGGGAAC)	DVKL^135^ALELGNLSI…^286^DLWQWRKSL^294^

## Data Availability

No new data were created or analyzed in this study. Data sharing is not applicable to this article.

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
