# Peer review of "Could Targeting NPM1c+ Misfolding Be a Promising Strategy for Combating Acute Myeloid Leukemia?"

_ijms, 2024, doi:10.3390/ijms25020811_

Round 1
Reviewer 1 Report
Comments and Suggestions for Authors
In this review Florio and Marasco discuss targeting a frequently mutated gene NPM1c+ as therapeutic strategy in AML. Overall it is a very import topic and the authors have covert the key content and literature. However, the authors should work on the structure of the review to make it easier to follow for the readers.
Detailed comments below:
“As a consequence of point mutations, NPM1 15 is unable to correctly fold and loses nuclear localization signal bearing an aberrant cytoplasmic localization, denoted as NPMc+."
While point mutations occur in NPM1 the key AML disease driving mutation is actually a insertion at W288 ranging from 2-12 added amino acids leading to a frame shift e.g. W288Cfs*12/Ffs*12/Lfs*12 https://www.oncokb.org/gene/NPM1/W288Cfs*12. The sentence above should be amended accordingly. Also throughout the manuscript NPM1c+ or NPMc+ the user should pick one and be consistent throughout the manuscript.
“Indeed, there are various subgroups of AML exhibiting i) t(8; 21) translocations and inv(16) abnormalities, ii) complex karyotypes, iii)TP53 mutations, iv)Mixed Lineage Leukemia gene rearrangements (MLL-PTD), v)FLT3-ITD mutations and vi)secondary AML, each characterized by specific genetic mutations, cytogenetic abnormalities and clinical features.”
I don’t fully understand the relevance of listing those genetic aberrations. They are many more. I suggest removing this sentence and adding a sentence that the genetic heterogeneity predicts outcome after standard chemotherapy (Papaemmanuil et al., NJEM 2016) and that AML genetics are now routinely used to tailor patient care (Doehner et al Blood 2022; 2022 ELN)
Also to introduce is that NPM1c outcome and response to therapy is highly dependent on co-occurring mutations and age (Straube et al Blood 2017; Papaemmanuil et al., NJEM 2016).
“NPM1 is an abundant multifunctional protein belonging to the nucleoplasmin family 55 of nuclear chaperones [7], present in high amounts in the granular region of nucleoli, 56 where takes part to rRNA maturation processes [8] and is essential for embryonic devel-57 opment [9], ribosome biogenesis [10], maintenance of genome stability [10], nucleolar 58 stress response [11], modulation of the p53-pathway [12] and regulation of apoptosis [13]. 59 It also plays a role in regulating gene transcription by interacting with specific factors and 60 directly binding to promoter regions [14]. NPM1 is observed in different cellular compart-61 ments in response to various types of cellular stress [15,16] and found overexpressed or 62 mutated in various types of tumors, including gastric, ovarian, bladder and prostate car-63 cinomas, as well as various hematological malignancies [17,18].”
This is a review about targeting NPM1c in AML. The authors should focus or highlight which of the functional role does NPM1 plays in hematopoiesis and AML and complement the findings with other cell types and processes.
“Figure 1. Structural and functional domains of NPM1 protein.”
I recommend not using abbreviations in the Figure 1 or at least spell out all abbreviations in the Figure 1 legend.
“In several cases of NPMc+ high expressions of the homeobox (HOX) gene and its 179 transcriptional cofactors MEIS1 (Myeloid Ecotropic Viral Integration Site 1) and PBX3 180 (Pre-B-Cell Leukemia Homeobox 3) [49] were encountered with increase of self-renewal 181 of leukemic clones [45], for this there is a growing interest on novel agents targeting XPO1 182 and HOX, which could affect the intranuclear re-localization of NPMc+ [50].”
A key characteristic of NPM1c+ in AML patient is the high expressions of the homeobox (HOX) transcription factors (TFs) including the HOXA/B cluster genes, MEIS1 (Myeloid Ecotropic Viral Integration Site 1) and PBX3 (Pre-B-Cell Leukemia Homeobox 3) [49]. Activation of these TF leads to an increase in self-renewal of leukemic clones [45], therefore inhibiting the expression of these TFs through menin-inhibitors or targeting XPO1 to prohibit NPM1c+ translocation are currently in clinical investigation (Clinical Studies)
NPMc+ => NPM1c+
AML mutations: key drivers of cellular mislocalization NPM1
This section in very long and I recommend sub sectioning it to make it easier to read and more structured
e.g.
3. NPM1 mutations in AML and its structural consequences
3.1 Common mutations NPM1c
3.2 Rare mutations Exon5
3.3 Fusion of NPM1c
4. Consequences of NPM1c mutations in AML
Cytoplasmic mislocation leading to
Hox expression activation..
Block in differentiation
Similar section 5 should be sub sectioned
5. Therapeutic strategies targeting NPMc+
Again I would subsection this paragraph
5.1 Therapeutics targeting protein-protein interaction
5.2 Therapeutics leading to NPM1 degradation
5.3 Therapeutics targeting NPM1c nuclear export
5.3 Therapeutics targeting of NPM1c targets
- menin inhibitors
It would be good to know and highlight if any of those compounds are in clinical trials and what phase.
Comments on the Quality of English Language
I recommend professional English language editing to improve readability.
Author Response
Reviewer: 1
In this review Florio and Marasco discuss targeting a frequently mutated gene NPM1c+ as therapeutic strategy in AML. Overall it is a very import topic and the authors have covert the key content and literature. However, the authors should work on the structure of the review to make it easier to follow for the readers.
Detailed comments below:
Q1: “As a consequence of point mutations, NPM1 is unable to correctly fold and loses nuclear localization signal bearing an aberrant cytoplasmic localization, denoted as NPMc+."
While point mutations occur in NPM1 the key AML disease driving mutation is actually a insertion at W288 ranging from 2-12 added amino acids leading to a frame shift e.g. W288Cfs*12/Ffs*12/Lfs*12 https://www.oncokb.org/gene/NPM1/W288Cfs*12. The sentence above should be amended accordingly. Also throughout the manuscript NPM1c+ or NPMc+ the user should pick one and be consistent throughout the manuscript.
A1: We thank the reviewer for the suggestion we have corrected accordingly, as follows:
“The most important mutation is the insertion at W288 which determines the frame shift W288Cfs12/Ffs12/Lfs*12 and leads to the addition of 2-12 amino acids which hamper the correct folding of NPM1. This mutation leads to the loss of the nuclear localization signal and to an aberrant cytoplasmic localization, denoted as NPM1c+”.
Q2: “Indeed, there are various subgroups of AML exhibiting i) t(8; 21) translocations and inv(16) abnormalities, ii) complex karyotypes, iii)TP53 mutations, iv)Mixed Lineage Leukemia gene rearrangements (MLL-PTD), v)FLT3-ITD mutations and vi)secondary AML, each characterized by specific genetic mutations, cytogenetic abnormalities and clinical features.”
I don’t fully understand the relevance of listing those genetic aberrations. They are many more. I suggest removing this sentence and adding a sentence that the genetic heterogeneity predicts outcome after standard chemotherapy (Papaemmanuil et al., NJEM 2016) and that AML genetics are now routinely used to tailor patient care (Doehner et al Blood 2022; 2022 ELN)
Also to introduce is that NPM1c outcome and response to therapy is highly dependent on co-occurring mutations and age (Straube et al Blood 2017; Papaemmanuil et al., NJEM 2016).
A2: We thank the reviewer for the comments and we agree. We have removed the sentence as suggested and added new sentences as follows:
“Genetic heterogeneity is a critical factor in predicting treatment response and developing therapeutic strategies [1]. Actually to tailor patients care, concomitant genetic mutations are also delineated, pointing out the importance of individualized assessment to optimize the efficacy of treatment protocols [2] [3].”
Q3: “NPM1 is an abundant multifunctional protein belonging to the nucleoplasmin family of nuclear chaperones [7], present in high amounts in the granular region of nucleoli, where takes part to rRNA maturation processes [8] and is essential for embryonic development [9], ribosome biogenesis [10], maintenance of genome stability [10], nucleolarstress response [11], modulation of the p53-pathway [12] and regulation of apoptosis [13].It also plays a role in regulating gene transcription by interacting with specific factors and directly binding to promoter regions [14]. NPM1 is observed in different cellular compartments in response to various types of cellular stress [15,16] and found overexpressed or mutated in various types of tumors, including gastric, ovarian, bladder and prostate car-cinomas, as well as various hematological malignancies [17,18].”
This is a review about targeting NPM1c in AML. The authors should focus or highlight which of the functional role does NPM1 plays in hematopoiesis and AML and complement the findings with other cell types and processes.
A3: We thank the reviewer for these suggestions. We have readapted the paragraph including the role that NPM1 plays in hematopoiesis in the revised manuscript as follows:
“NPM1 is an multifunctional protein with nuclear chaperone functions [4], mainly present in nucleoli, where takes part to rRNA maturation processes [5] and embryonic development [6]. It is directly implicated in primitive hematopoiesis and hematopoietic malignancies as myelodysplastic syndrome (MDS) since NPM1 plays a critical role in the maintenance of hematopoietic stem cells (HSCs) in preserving the functional integrity of these cells in the context of competitive transplantation (HSCT) [7] and the transformation of MDS into leukemia as well as in the modulation of gene expression and signaling pathways that govern cell survival [8]. NPM1 is observed in different cellular compartments in response to various types of cellular stress [9,10] and found overexpressed or mutated in various types of tumors, including gastric, ovarian, bladder and prostate carcinomas and in hematological malignancies [11,12].”
Q4: “Figure 1. Structural and functional domains of NPM1 protein.”
I recommend not using abbreviations in the Figure 1 or at least spell out all abbreviations in the Figure 1 legend.
A4: We thank the reviewer for the suggestion. We have revised the caption of Figure 1, explicating abbreviations.
Q5: “In several cases of NPMc+ high expressions of the homeobox (HOX) gene and its transcriptional cofactors MEIS1 (Myeloid Ecotropic Viral Integration Site 1) and PBX3 (Pre-B-Cell Leukemia Homeobox 3) [49] were encountered with increase of self-renewal of leukemic clones [45], for this there is a growing interest on novel agents targeting XPO1 and HOX, which could affect the intranuclear re-localization of NPMc+ [50].”
A key characteristic of NPM1c+ in AML patient is the high expressions of the homeobox (HOX) transcription factors (TFs) including the HOXA/B cluster genes, MEIS1 (Myeloid Ecotropic Viral Integration Site 1) and PBX3 (Pre-B-Cell Leukemia Homeobox 3) [49]. Activation of these TF leads to an increase in self-renewal of leukemic clones [45], therefore inhibiting the expression of these TFs through menin-inhibitors or targeting XPO1 to prohibit NPM1c+ translocation are currently in clinical investigation (Clinical Studies).
A5: We thank the reviewer for the valuable feedback. We promptly edited the sentence as suggested. In addition, we have reported the NCT number (National Clinical Trial), from ClinicalTrials.gov, for ongoing clinical trials of menin inhibitors or agents targeting XPO1.
Q6: NPMc+ => NPM1c+
Done
Q7: AML mutations: key drivers of cellular mislocalization NPM1
This section in very long and I recommend sub sectioning it to make it easier to read and more structured.
e.g.
- NPM1 mutations in AML and its structural consequences
3.1 Common mutations NPM1c
3.2 Rare mutations Exon5
3.3 Fusion of NPM1c
3.4. Consequences of NPM1c mutations in AML
3.5 Cytoplasmic mislocation leading to
3.6 Hox expression activation..
3.7 Block in differentiation
A7: We thank the reviewer for these suggestions and we agree to sub-section paragraph 3 on AML mutations and taking into account similar lengths of sub-sections we carried the following division:
- NPM1 is the most commonly mutated gene in adult AML
- 1 Common NPM1 mutations and structural consequences
- 2 Rare mutations of exon 5 and fusion transcripts of NPM1c+
- 3 Consequences of cytoplasmic mislocation of NPM1c+
Q8: Similar section 5 should be sub sectioned
- Therapeutic strategies targeting NPMc+
Again I would subsection this paragraph
5.1 Therapeutics targeting protein-protein interaction
5.2 Therapeutics leading to NPM1 degradation
5.3 Therapeutics targeting NPM1c nuclear export
5.3 Therapeutics targeting of NPM1c targets
- menin inhibitors
It would be good to know and highlight if any of those compounds are in clinical trials and what phase.
A8: We appreciate the reviewer's suggestion and agree that this modification may allow a more focused and organized exploration of the discussed molecules. Accordingly, we have divided section 5 into sub-sections based on the target. In addition, as suggested, we introduced a new sub-section on Menin inhibitors in NPM1c+ -AML and currently in clinical trials (reporting the NCT number) as also suggested by reviewer 2.
So, we organized new sub-sections as follows:
5.1 Therapeutics targeting NPM1 protein-protein interactions.
5.2 Therapeutics targeting the nucleolus of NPM1c+.
5.3 Therapeutics targeting NPM1c+ localization
5.4 Therapeutics as Menin inhibitors
5.5 Therapeutics targeting aggregation of NPM1c+.
Furthermore, we have incorporated a new figure (Figure 5), showcasing the chemical structures of Menin inhibitors presently in clinical trials. For those not currently in clinical trials, we have modified the existing Figure 4.
Figure 5. Chemical structures of Menin inhibitors currently undergoing clinical evaluation.

Reviewer 2 Report
Comments and Suggestions for Authors
The review article of Florio et al. gives an overview on the biology of NPM1c mutated AML detailing the structural and functional consequences of these mutations and details several novel treatment approaches.
Overall the article is well and clearly written and summarizes in a comprehinsiveway almost all recent data of the topic.
Some minor issues should be addressed:
1. Recent data provides promising insights for the use of menin inhibitors in NPM1c mutated AML with menin inhibitors entering clinical trials. This should be included in the therapeutic strategies section.
2. NPM1, NPM1, NPMc+ etc. should be homogenized all over the paper for gene and protein respectively
3. Pp1, ll 31, allogenic stem cell transplantation should be added.
4. Pp 7, ll 290, red bracket should be black
Author Response
Reviewer 2
The review article of Florio et al. gives an overview on the biology of NPM1c mutated AML detailing the structural and functional consequences of these mutations and details several novel treatment approaches.
Overall, the article is well and clearly written and summarizes in a comprehensive way almost all recent data of the topic.
We thank the reviewer for his/her positive comment.
Some minor issues should be addressed:
- Recent data provides promising insights for the use of menin inhibitors in NPM1c mutated AML with menin inhibitors entering clinical trials. This should be included in the therapeutic strategies section.
We appreciate the reviewer's suggestion and have duly considered it. In response, we have incorporated a dedicated section on therapeutic strategies focusing on menin inhibitors, emphasizing their entry into clinical trials. Also, for their chemical structures, we included a new figure (Figure 5).
- NPM1, NPM1, NPMc+ etc. should be homogenized all over the paper for gene and protein respectively
Done
- Pp1, ll 31, allogenic stem cell transplantation should be added.
Done. We added the allogenic stem cell transplantation as follows:
“Acute myeloid leukemia (AML) is a prevalent form of acute leukemia in adults, with a cure rate of 40-50% among individuals aged 18-60 years when treated with standard chemotherapy and/or allogeneic stem cell transplantation.”
- Pp 7, ll 290, red bracket should be black
Done

Round 2
Reviewer 1 Report
Comments and Suggestions for Authors
The authors have addressed all my concerns/suggestions.